SciPost Physics

Submission

# Uncertainty in mean $X_{\max}$ from diffractive dissociation estimated using measurements of accelerator experiments

Ken Ohashi[1*], Hiroaki Menjo[1], Takashi Sako[2], and Yoshitaka Itow[1,3]

**1** Institute for Space-Earth Environmental Research, Nagoya University
**2** Institute for Cosmic Ray Research, the University of Tokyo
**3** Kobayashi-Maskawa Institute for the Origin of Particles and the Universe, Nagoya University
* ohashi.ken@isee.nagoya-u.ac.jp

October 6, 2022

## Abstract

**Mass composition is important for understanding the origin of ultra-high-energy cosmic rays. However, interpretation of mass composition from air shower experiments is challenging, owing to significant uncertainty in hadronic interaction models adopted in air shower simulation. A particular source of uncertainty is diffractive dissociation, as its measurements in accelerator experiments demonstrated significant systematic uncertainty. In this research, we estimate the uncertainty in $\langle X_{\max} \rangle$ from the uncertainty of the measurement of diffractive dissociation by the ALICE experiment. The maximum uncertainty size of the entire air shower was estimated to be $^{+4.0}_{-5.6}$ g/cm$^2$ for air showers induced by $10^{17}$ eV proton, which is not negligible in the uncertainty of $\langle X_{\max} \rangle$ predictions.**

## 1 Introduction

Mass composition of ultra-high energy cosmic rays is important for understanding the origin of these cosmic rays, as acceleration at the source and interactions during propagation depends on composition; acceleration of such cosmic rays depends on their charge if we assume acceleration by magnetic fields. Interactions with cosmic-microwave background photons during propagation vary between nuclei. Measurements of these cosmic rays have been performed using observations of air showers induced by cosmic rays, for example, the Telescope Array experiment [1] and the Pierre Auger Observatory [2]. The depth of maximum of air shower developments, $X_{\max}$, are widely measured as an estimator of mass composition. Mass composition is interpreted by comparing the measurements of $X_{\max}$ and its predictions using simulation. However, simulation predictions vary if the hadronic interaction model adopted changes to another model. Precise understanding of hadronic interactions is crucial for mass composition interpretation.

Hadronic interaction models were updated using accelerator experiments. For example, EPOS-LHC [3, 4] were tuned using measurements of inelastic cross sections, the distribution of charged

particles, and particle productions by experiments at the Large Hadron Collider (LHC). Meanwhile, measurements of diffractive dissociation by experiments at LHC have significant uncertainty [5,6]. The effects of diffractive dissociation were discussed in our previous study [7],where differences in the cross sections of diffractive dissociation among hadronic interaction models affected 8.9 g/cm$^2$ on $\langle X_{max} \rangle$ for air showers induced by $10^{19}$ eV protons. Uncertainty in diffractive dissociation measurements can affect $\langle X_{max} \rangle$. In this research, we estimate the effects of uncertainty in the measurements on $\langle X_{max} \rangle$ for air showers induced by $10^{17}$ eV protons. After categorizing the simulated events using the definitions in the ALICE experiment, we weighted the fractions of each category by the ratio of the experimental data to the predictions.

## 2 Diffractive dissociation

Diffractive dissociation is one type of hadronic interaction, caused by the exchange of a pomeron and characterized by low-momentum transfer. In the collision, a colliding particle is scattered, becomes a diffractively excited state, and subsequently dissociates into particles. The other colliding particle can either be intact or dissociate. If the other colliding particle remains intact, the collisions are called single diffractions (SD). If both colliding particles dissociate, the collisions are called double diffractions (DD). From a cosmic-ray point of view, four types of diffractive dissociation exist: single diffraction with projectile cosmic-ray dissociation (projectile SD), single diffraction with target air nucleus dissociation (target SD), double diffraction (DD), and central diffraction (CD), in which two colliding particles were intact, however, particles were produced in the exchange of two or more Pomerons. Hereafter, collision types other than these types in the hadronic interaction are considered non-diffractive collisions (ND). Notably, CD are not considered and included in non-diffractive collisions in this study, as some models predict extremely small cross sections for CD.

## 3 Air shower simulation

In this study, air showers were simulated using the air shower simulation package CONEX v6.40 [8]. EPOS-LHC [3, 4] and SIBYLL 2.3c [9, 10] were adopted as hadronic interaction models for collisions induced by particles above 80 GeV. UrQMD [11, 12] was adopted as a hadronic interaction model for low-energy collisions. Two samples were simulated; 40000 showers, hereafter sample a), were simulated using EPOS-LHC and SIBYLL 2.3c, respectively. Additionally, air showers with projectile SD, target SD, and DD at the first interaction were simulated by changing simulation codes. Hereafter this sample is referred to as sample b). 1000 showers were simulated for each case in sample b).

These simulated samples were categorized by collision type, diffractive mass, and rapidity gap. The collision type was defined using type information in each hadronic interaction model. For EPOS-LHC, the collision type information in the model was used. For SIBYLL 2.3c, the type information was provided for each interaction between two partons. If an interaction consists of one interaction between two partons and classified as diffractive dissociation, the collision was considered diffractive dissociation. The dissociation system was separated using the largest rapidity gap considering all particles for DD and target SD and using a threshold to separate the dissociation system for projectile SD. The threshold rapidity gap was set at 1.5 in the laboratory system. If,

by accident, only one particle in the dissociation system exists in the process of the identification of dissociation system, the second largest rapidity gap was adopted to separate the dissociation system. The diffractive mass was subsequently calculated from the momentum of particles in the dissociation system. Gaps between charged particles in pseudorapidity were calculated from the distribution of produced charged particles and sorted by the pseudorapidity of each particle. The largest gap was considered the rapidity gap $\Delta\eta$. Collision types were added to the outputs for both samples a) and b). Rapidity gaps and the diffractive mass were only calculated in sample b) to consider definitions of the experimental result. Notably, sample a) was identical to the samples used in [7].

# 4 Analysis method

In this work, we focus on the effects of the first intercation of air showers. We categorize simulated air showers by collision types at the first intercation and the mean value was calculated. The mean value of $X_{\max}$, $\langle X_{\max}^{\text{all}} \rangle$, of each categorized sample was calculated as follows:

$$\langle X_{\max}^{\text{all}} \rangle = \sum^{i} f^{i} \langle X_{\max}^{i} \rangle, \tag{1}$$

where $i$ runs over all categorized samples. $f^{i}$ is the fraction of each category in the total sample. By changing the fraction $f^{i}$ in Eq. 1, we estimated the effect of each fraction.

We modified the fractions based on the LHC experimental result. Using cross sections of SD and DD from MC simulation, $\sigma_{\text{MC}}^{i}$, where $i$ runs for SD and DD, and the experimental result of cross sections, $\sigma_{\text{Data}}^{i}$, the ratio of experimental data to predictions by the simulation, $R_{\text{Data/MC}}^{i} = \sigma_{\text{Data}}^{i}/\sigma_{\text{MC}}^{i}$, was calculated for each category. The ratios were applied to modify fractions. The modified $\langle X_{\max} \rangle$ was then calculated using modified fractions and Eq. 1. Using the uncertainty of experimental data, the uncertainty of $R_{\text{Data/MC}}$ and finally $\langle X_{\max} \rangle$ can be calculated. We note that the inelastic cross sections remain unchanged. The effects of differences in particle productions of diffractive dissociation were not considered, while they demonstrated minor effects on $\langle X_{\max} \rangle$ [7].

We consider a measurement of single and double diffraction by the ALICE experiment for proton-proton collisions with $\sqrt{s} = 7$ TeV [5]. The cross section of single diffraction $\sigma^{\text{SD}}$ and double diffraction $\sigma^{\text{DD}}$ measured by the ALICE experiment was $14.9^{+3.4}_{-5.9}$ mb and $9.0 \pm 2.6$ mb, respectively. $\sigma^{\text{SD}}$ was measured for $M_X < 200$ GeV/c$^2$, where $M_X$ was the diffractive mass of the dissociation system. $\sigma^{\text{DD}}$ was measured for $\Delta\eta > 3$. We note that $\Delta\eta$ was the pseudo-rapidity gap for charged particles and non-diffractive collisions were not subtracted in the measurement. $R_{\text{Data/MC}}$ were calculated from the experimental result and simulations of EPOS-LHC and SIBYLL 2.3 by CRMC v1.6 [13]. $R_{\text{Data/MC}}$ for EPOS-LHC was $1.95^{+0.45}_{-0.78}$ for single diffraction and $0.54^{+0.16}_{-0.16}$ for double diffraction. $R_{\text{Data/MC}}$ for SIBYLL 2.3 was $1.85^{+0.43}_{-0.73}$ for single diffraction and $0.38^{+0.11}_{-0.11}$ for double diffraction. Then, to calculate the modified $\langle X_{\max} \rangle$ and its uncertainty, these $R_{\text{Data/MC}}$ calculated from the proton-proton collision were applied for the first proton-air nucleus interaction in an air shower with two assumptions; one was that the $R_{\text{Data/MC}}$ calculated at $\sqrt{s} = 7$ TeV can be applied for collisions induced by the $10^{17}$ eV proton, although the center-of-mass energy differed slightly. The other was that $R_{\text{Data/MC}}$ calculated from *proton-proton collisions* can be applied for predictions of *proton-air nucleus collisions*. Considering the second assumption, we rely on proton-nucleus collision modeling in each hadronic interaction model, therefore results differences in the modified $\langle X_{\max} \rangle$ owing to hadronic interaction models were expected. We note that differences

| interaction model | collision type in the model | categorized by the ALICE definitons | | | | |
|---|---|---|---|---|---|---|
| | | the number of events | | | $\langle X_{\max} \rangle$ [g/cm$^2$] | |
| | | total | diffraction | non-diffraction | diffraction | non-diffraction |
| EPOS-LHC | projectile SD | 1000 | 502 | 498 | 732.33 ± 0.14 | 722.10 ± 0.13 |
| | target SD | 1000 | 609 | 391 | 735.51 ± 0.12 | 720.59 ± 0.18 |
| | DD | 1000 | 647 | 353 | 731.56 ± 0.10 | 711.56 ± 0.17 |
| | ND | 10000 | 973 | 9027 | 714.91 ± 0.07 | 684.12 ± 0.01 |
| SIBYLL 2.3c | projectile SD | 1000 | 643 | 357 | 729.30 ± 0.11 | 729.94 ± 0.20 |
| | target SD | 1000 | 638 | 362 | 755.72 ± 0.13 | 749.42 ± 0.23 |
| | DD | 1000 | 746 | 254 | 725.38 ± 0.09 | 722.68 ± 0.25 |
| | ND | 10000 | 2557 | 7443 | 723.41 ± 0.03 | 693.50 ± 0.01 |

Table 1: The number of events and $\langle X_{\max} \rangle$ of sample b) with categorization using the definitions in the ALICE experiment [5]. 1000 or 10000 showers were simulated for each collision type at the first interaction in each model.

| | Projectile SD | Target SD | DD (including ND) | others |
|---|---|---|---|---|
| fraction [%] | 2.0 | 2.7 | 13.1 | 82.2 |
| $\langle X_{\max} \rangle$ [g/cm$^2$] | 732.3 | 735.5 | 721.5 | 688.0 |

Table 2: Fractions and $\langle X_{\max} \rangle$ categorized at the first proton-air interaction of air showers by following definitions of the ALICE experiment. EPOS-LHC was adopted as a hadronic interaction model for high energy.

between SIBYLL 2.3 and SIBYLL 2.3c were ignored in this study, as these differences were relevant to particle productions in fragmentation and beam remnants, not for diffractive dissociation [10].

# 5 Result and discussions

## 5.1 Simulation results of fractions and $\langle X_{\max} \rangle$ with categorization using the ALICE experiment definitions

Table 1 shows the fractions and $\langle X_{\max} \rangle$ considering the definitions in the ALICE experiment result [5] for sample b). 1000 or 10000 air showers were simulated for each collision type at the first interaction based on the definition in each hadronic interaction model. These were subsequently classified into diffraction and non-diffraction based on the experiment definitions [5]. We note that the definitions for the result of SD in the ALICE experiment were considered for projectile SD and target SD, and that of DD were considered for DD and ND. Finally, the fraction and $\langle X_{\max} \rangle$ of air showers categorized by definitions in [5] were calculated using the results of sample a), which are summarized in [7], and the results in Table 1. The results are shown in Tables 2 and 3.

| | Projectile SD | Target SD | DD (including ND) | others |
|---|---|---|---|---|
| fraction [%] | 4.3 | 1.9 | 23.5 | 70.3 |

Table 3: Fractions of air showers at the first proton-air intercation are categorized by following the definition of the ALICE experiment. SIBYLL 2.3c was adopted as a hadronic interaction model for high energy.

## 5.2   Results of the modified $\langle X_{\max} \rangle$ and its uncertainty

The modified $\langle X_{\max} \rangle$ and its uncertainty were calculated using the method described in Section 4 and the fractions and $\langle X_{\max} \rangle$ in Tables 2 and 3. The results were $694.6^{+1.2}_{-1.8}$ g/cm$^2$ using fractions predicted by EPOS-LHC and $696.2^{+1.5}_{-2.2}$ g/cm$^2$ using fractions predicted by SIBYLL 2.3c. For both results, $\langle X_{\max} \rangle$ simulated with EPOS-LHC was adopted. The difference between the two modified $\langle X_{\max} \rangle$, which was 1.6 g/cm$^2$, stemmed from treatments of proton-nucleus collisions in each hadronic interaction model. Total uncertainty for the first interaction considering the result of the ALICE experiment was $^{+1.7}_{-2.3}$ g/cm$^2$ calculated from the 1.6 g/cm$^2$ difference between two models and larger uncertainty in the two modified results, which was $^{+1.5}_{-2.2}$ g/cm$^2$.

This estimation only considers the effects of diffractive dissociation at the first interaction. In our previous study [7], we estimated the ratio of the effect of the entire air shower to the effect at the first interaction, which was a maximum of 2.4. Thus, the maximum size of uncertainty from the result of the ALICE experiment for the entire air shower is estimated to be $^{+4.0}_{-5.6}$ g/cm$^2$ by multiplying $^{+1.7}_{-2.3}$ g/cm$^2$ by a factor 2.4 [7]. This size of uncertainty corresponds to approximately half of the difference in $\langle X_{\max} \rangle$ predictions among hadronic interaction models. Although the difference is caused by several sources [14], half of the difference is not negligible in the uncertainty of $\langle X_{\max} \rangle$ predictions.

## 6   Conclusion

In this research, the effects of uncertainty in the accelerator experiment on $\langle X_{\max} \rangle$ were estimated. Concentrating on the first interaction of air showers, the uncertainty of $\langle X_{\max} \rangle$ owing to the uncertainty in the diffractive dissociation measurements by the ALICE experiment [5] was estimated to be $^{+1.7}_{-2.3}$ g/cm$^2$. The maximum size of uncertainty of the entire air shower was estimated to be $^{+4.0}_{-5.6}$ g/cm$^2$, which is not negligible for the uncertainty of $\langle X_{\max} \rangle$ predictions.

## Acknowledgements

**Funding information**   K.O. was supported by Grant-in-Aid for JSPS Fellows (JP21J11122).

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
