# Peer review of "Uncertainty in mean $X_{\rm max}$ from diffractive dissociation estimated using measurements of accelerator experiments"

_SciPost Physics Proceedings_

## Round 1 · Referee Report · Yasushi Muraki (Referee 1) · 2022-8-24

Report
I have once again read this paper and checked the numbers given in section 4. Espexcially the numbers given in line 7 to 9 from the bottom of page 3. The number seems to be inconsistent each other. I recomend to the authors, once again check the numbers.
single diffractive double diffractive
ALICE experiment 14.9mb 9.0mb
EPOS prediction 3.4mb 9.6 mb
SYBILL prediction 4.5mb 17.2mb according Ohash Sigma inelastic collision 73.2 mb Then the ratio R_data/MC
EPOS 0.23 0.94
SIBYLL 0.30 0.52
Ohashi provided the nubers in page 3 line 7-9 from the bottom
EPOS 1.95 0.54
SIBYLL 1.85 0.38
After fixing this problem, the paper is worthwhile to be published.
Requested changes
page 2 line 8, ALICE should be included-->
using the definitions in the ALICE experiment,
page 4 line 4 from the boottom , ALICE should be uncluded.
the result of SD in the ALICE experiment
Author: Ken Ohashi on 2022-09-27 [id 2856]
(in reply to Report 1 by Yasushi Muraki on 2022-08-24)
Dear Muraki-san,
Thank you for your comments.
We updated the manuscripts.
Here are the answers to your questions and comments.
I have once again read this paper and checked the numbers given in section 4. Espexcially the numbers given in line 7 to 9 from the bottom of page 3. The number seems to be inconsistent each other. I recomend to the authors, once again check the numbers.
single diffractive double diffractive
ALICE experiment 14.9mb 9.0mb
EPOS prediction 3.4mb 9.6 mb
SYBILL prediction 4.5mb 17.2mb according Ohash Sigma inelastic collision 73.2 mb Then the ratio R_data/MC
EPOS 0.23 0.94
SIBYLL 0.30 0.52
Ohashi provided the nubers in page 3 line 7-9 from the bottom
EPOS 1.95 0.54
SIBYLL 1.85 0.38
After fixing this problem, the paper is worthwhile to be published.
Answer:
The ALICE paper reported cross-sections for proton-proton collisions, while numbers in table 3 showing for proton air nucleus collisions.
We first calculated the ratio RData/MC from the proron-proton collision, then we applied it to the proton-air nucleus collision. We assume the proton-nucleus collision modeling in EPOS-LHC and SIBYLL 2.3. Differences of modeling between EPOS-LHC and SIBYLL 2.3 were considered in the uncertainty of <X_{\rm max} >.
Therefore, the numbers in table 3 were calculated by applying the ratio RData/MC to cross-sections of proton-air nucleus collisions.
We revised the descriptions in Section 4 to make this point clear.
Requested changes
page 2 line 8, ALICE should be included-->
using the definitions in the ALICE experiment,
-> Answer: we applied this modification.
page 4 line 4 from the boottom , ALICE should be uncluded.
the result of SD in the ALICE experiment
-> Answer: we applied this modification.
Anonymous on 2022-08-19 [id 2739]
The discussions provided in Section 5.1 of page 4 provided a confusion.
According to the paper of Alice group
(Eur Phys. J.C (2013) 73:2456)(Your reference numbre is [4]),
The single diffrcative process to ineleastic cross section
is estimated as 0.20 and DD is estimated as 0.12.
On the other hand, you provided the related numbers in Table 3
as 4.3+1.9 and 23.5. For the single diffractive collsion process,
your estimation is half , while for DD process your value is twice high.
Please give appropriate explanation on this point.
*) The Tbale 3 captiions, is better to write clearly as;
Fractions of air showers at the first intercation are categorized
by following the definition of the ALICE experiment.
*) page 1 reference [1]. The Auger project should be cited.
*) page 2 the last line. The threshold rapidity gap was set at 1.5
in the laboratory system.
Anonymous on 2022-09-27 [id 2855]
(in reply to Anonymous Comment on 2022-08-19 [id 2739])Thank you for your comments.
The discussions provided in Section 5.1 of page 4 provided a confusion.
According to the paper of Alice group
(Eur Phys. J.C (2013) 73:2456)(Your reference numbre is [4]),
The single diffrcative process to ineleastic cross section
is estimated as 0.20 and DD is estimated as 0.12.
On the other hand, you provided the related numbers in Table 3
as 4.3+1.9 and 23.5. For the single diffractive collsion process,
your estimation is half , while for DD process your value is twice high.
Please give appropriate explanation on this point.
Answer :
The ALICE paper reported cross-sections for proton-proton collisions, while numbers in table 3 showing for proton air nucleus collisions.
We first calculated the ratio RData/MC from the proron-proton collision, then we applied it to the proton-air nucleus collision. We assume the proton-nucleus collision modeling in EPOS-LHC and SIBYLL 2.3. Differences of modeling between EPOS-LHC and SIBYLL 2.3 were considered in the uncertainty of <X_{\rm max} >.
We revised the descriptions in Section 4 to make this point clear.
*) The Tbale 3 captiions, is better to write clearly as;
Fractions of air showers at the first intercation are categorized
by following the definition of the ALICE experiment.
-> Answer: we applied this modification.
*) page 1 reference [1]. The Auger project should be cited.
-> Answer: we applied this modification.
*) page 2 the last line. The threshold rapidity gap was set at 1.5
in the laboratory system.
-> Answer: we applied this modification.

---

## Round 2 · Referee Report · Anonymous (Referee 2) · 2022-10-1

Report

This draft has been improved and the paper should be publlished in the poceeding, but after a few changes of the text.

Requested changes

(1) page 3 just after the chapter 4, Analysis method. A few lines should be written correctly. for example,
In this work, we focus on the effects of the first intercation of air showers. We categorize simulated air showers by collision types at the first intercation and the mean value was calculated. The mean value of Xmax < >, of each categorized samples was calculated as following definition of f and Eq.(1)where i..samples. By changing the fraction f in Eq.1, we have estimated the effect of each fraction.
(2) page 5 line 8 from the top. Total uncertainty for the first intercation considering the result of the ALICE experiment was .. calculated from the 1.6g/cm2 difference bewteen the two models and the uncertainty in the two....
(3) page 5 line 12 from the top. You may need the reference where you have provided the number 2.4.
(4) page 5 line 15. As the difference --> Although the difference

---

## Round 2 · Author Response

We revised manuscripts considering all comments by the referee.

---

## Round 2 · List of Changes

Update descriptions in section 4 to avoid confusion.
Add one reference and apply several modifications suggested by the referee.

---

## Round 3 · Author Response

We applied suggested modifications.

---

## Round 3 · List of Changes

We applied all modifications suggested by the reviewer.

---

## Editorial Decision

editorial_decision: